# Human Flow Dataset Reveals Changes in Citizens’ Outing Behaviors including Greenspace Visits before and during the First Wave of the COVID-19 Pandemic in Kanazawa, Japan

**DOI:** 10.3390/ijerph19148728

**Published:** 2022-07-18

**Authors:** Yusuke Ueno, Sadahisa Kato, Tomoka Mase, Yoji Funamoto, Keiichi Hasegawa

**Affiliations:** 1Faculty of Bioresources and Environmental Sciences, Ishikawa Prefectural University, Nonoichi 921-8836, Japan; uenoyu@ishikawa-pu.ac.jp (Y.U.); zhijiajianlai@gmail.com (T.M.); 2Faculty of Environmental Studies, Tottori University of Environmental Studies, Tottori 689-1111, Japan; 3Fukuyama Consultants Co., Ltd., Tokyo 101-0033, Japan; y.funamoto@fukuyamaconsul.co.jp; 4Social Value Incubation Lab., Tokyo 101-0033, Japan; 5EY Strategy and Consulting Co., Ltd., Tokyo 100-0006, Japan; keiichi.hasegawa2@jp.ey.com

**Keywords:** behavioral change, COVID-19 pandemic, urban greenspace, mental health, human flow

## Abstract

Greenspaces, including parks, provide various socio-ecological benefits such as for aesthetics, temperature remediation, biodiversity conservation, and outdoor recreation. The health benefits of urban greenspaces have received particular attention since the onset of the COVID-19 pandemic, which has triggered various movement restrictions and lifestyle changes, including regarding the frequency of people’s visits to greenspaces. Using mobile-tracking GPS data of Kanazawa citizens, we explored how citizens’ behaviors with respect to outings changed before and during Japan’s declaration of a COVID-19 state of emergency (April–May 2020). We also examined citizens’ greenspace visits in relation to their travel distance from home. We found that Kanazawa citizens avoided going out during the pandemic, with a decrease in the number, time, and distance of outings. As for the means of transportation, the percentage of outings by foot increased on both weekdays and holidays. While citizens refrained from going out, the percentage change of the percentage in large greenspace visits increased very slightly in 2020. As for greenspace visitation in 2020 compared to 2019, we found that citizens generally visited greenspaces closer to their homes, actually increasing visitation of nearby (within 1000 m) greenspaces. This study of how outing behaviors and greenspace use by Kanazawa citizens have changed underscores the value of nearby greenspaces for physical and mental health during movement restrictions under the pandemic.

## 1. Introduction

Urban greenspaces provide numerous socio-ecological benefits such as aesthetics, microclimate remediation, biodiversity conservation, environmental education, and recreation to urban residents [1,2,3,4,5]. The COVID-19 pandemic has also highlighted the important health benefits of urban greenspaces [6,7,8,9,10,11,12,13]. The role of urban greenspaces in maintaining and increasing people’s health [12,14], including physical, mental, and social health benefits [15,16,17,18,19,20], has become an important research theme. For example, a Danish survey indicates that access to a garden or proximity to green areas are associated with less stress and a lower likelihood of obesity [21]. An online questionnaire survey in Tokyo during the COVID-19 pandemic found that the frequency of greenspace use and even the presence of windows looking out onto greenery were positively associated with various mental health outcomes [22].

The COVID-19 pandemic has triggered movement restrictions and lifestyle changes, including people’s greenspace use. Particularly in Europe, the United States, and China, strict movement restrictions (e.g., stay-at-home orders and restrictions on commuting and traveling) were adopted, and the cities were “locked down”. Social (physical) distancing was enforced, and restaurant and bar business hours were shortened or closed.

Japan’s state of emergency declaration differs from the strict measures taken elsewhere. Then Prime Minister Shinzo Abe declared a state of emergency in seven prefectures, including Tokyo, Osaka, and Fukuoka, on 7 April 2020, and expanded its scope nationwide on 16 April [23]. This declaration empowered the prefectural governor to request that residents refrain from going out unnecessarily for a specified period and in a specified area and that businesses limit their operation hours. The emergency restrictions in Japan were voluntary (requested by the government), not mandatory, and no penalties or fines were imposed [23,24]. Nevertheless, most people and businesses complied.

The originality of our research rests in its use of human flow data (big data from GPS-equipped mobile phones) to analyze the changes in people’s greenspace visitation during the pandemic. Human flow data enable us to understand where, when, and how many people are present. Use of such data is expected to solve issues in such fields as tourism, transportation, disaster prevention, urban planning, and public health [25,26,27,28]. Human flow data can be collected through fixed-point monitoring, questionnaires and interviews, wireless terminals and GPS loggers, and device location information obtained from mobile phones and car navigation systems with built-in GPS/GNSS receivers [29]. The fourth type of human flow data enables a detailed analysis of behavioral changes by tracking movements of greenspace users [30,31]. Improved camera and sensor performance, as well as cellphone base station information and GPS information, have facilitated obtaining human flow big data.

Few studies have used human flow data (mobile phone location data) to examine people’s greenspace use [31,32,33]. Human flow data with their rich spatio-temporal information provide useful information for planning and designing post-COVID-19 urban greenspaces (e.g., [34]). Using a location data point, for example, Hu et al. [33] demonstrated the applicability of locational data to analyzing the population density distribution and activity trajectory in an urban park. Using human flow big data, we explored how citizens’ outing behaviors may have changed before and during the first wave of the COVID-19 pandemic in Kanazawa, Japan, in April–May 2020. We first examined overall outing and visitation patterns related to urban parks and other greenspaces, followed by citizens’ greenspace visits in relation to their travel distance from home. The pandemic has enabled a great natural experiment [35] in that the pre-COVID-19 world serves as a “business-as-usual” (BAU) model (in a reverse sense). Many subsequent societal and lifestyle changes may be irreversible. We compared 2020 data, in the middle of the first wave of the global COVID-19 pandemic, with 2019 data. This comparison highlights the stark difference between the BAU and the world we know.

## 2. Materials and Methods

### 2.1. Study Area

The study area, Kanazawa City in Ishikawa Prefecture, faces the Sea of Japan (Figure 1). It measures 468.79 km^2^ and has an estimated population of 462,814 (as of 1 April 2020). Surrounded by mountains, rivers, and the sea, Kanazawa City has 574 urban parks, covering 308.20 ha (as of 1 April 2016) [36]. Since Kanazawa was spared from war and other major disasters, many of the traditional townscapes from the feudal domain era remain and are a valuable asset to Kanazawa City, attracting many domestic and international tourists.

The first COVID-19 case was confirmed in Ishikawa Prefecture on 21 February 2020 [37]. To prevent the infection from spreading, all prefectural schools were temporarily closed; events were postponed or cancelled; facilities were closed; and residents were asked to refrain from going out unnecessarily and traveling to and from other prefectures. On 13 April, the prefecture issued its own emergency declaration, and on 16 April, the Japanese government named it one of the “designated prefectures on alert”, which required the prefecture to take special measures to prevent the spread of COVID-19, and the residents were asked to refrain from going out. The alert in Ishikawa continued until 14 May.

### 2.2. The Human Flow Data and Sample

The human flow data used in the analysis were the location data of cellphone terminals (mobile GPS data) collected by Blogwatcher Inc. (Tokyo, Japan) (location data service provider for smartphones) with individual consent. Data had been “generalized” so that individuals could not be identified. These data consisted of the latitude and longitude recorded by the built-in GPS of each terminal (with an error accuracy of about 50 to 100 m), the time of recording (every five minutes to several hours), and the estimated area of residence.

The data covered 15 days each, from 1 to 15 May 2019, and from 1 to 15 May 2020. In Japan, there are consecutive holidays in early May of every year, when many people tend to enjoy leisure and travel. From 1 to 15 May 2019, there were eight “holidays” (weekends and holidays) and seven weekdays; from 1 to 15 May 2020, there were seven “holidays” and eight weekdays. Since the aim of the study was to compare changes in people’s outing behaviors before and during the pandemic, and many people tend to go outside during the long holiday week in May, we hypothesized that large differences in outing behaviors could be observed in the selected study periods. Since the data included the records of those who were visiting from outside the city, from the data collected during these periods, we extracted the records that continuously stayed in Kanazawa City at night, and treated them as Kanazawa citizens. To enable comparison between 2019 and 2020, and to reduce the effect of variations in GPS data acquisition time between mobile devices, only devices with two years of records and data acquisition time of more than 20 h per day were selected for analysis. The total number of data points (terminals) used in this study was 13,710 for two years, and GPS logs numbered 3,542,408 in total. Of these, 5482 terminals were able to record data in both 2019 and 2020, equivalent to about 1.18% of Kanazawa citizens. This study was approved by the Research Ethics Committee of the first author’s institution before being conducted.

### 2.3. Analysis Methods

#### 2.3.1. Kanazawa Citizens’ Behavioral Changes

To analyze changes in the overall behavior of Kanazawa citizens before and during the COVID-19 pandemic, 2019 and 2020 data were compared regarding the number of outings per day, time spent outside the home, distance outside the home (straight-line distance from home to place of visit), means of transportation, and types (large green areas, commercial areas) and locations of outings. The home (i.e., the estimated area of residence) was presumed to be near the place where Kanazawa citizens mostly stayed overnight. The means of transportation was determined based on the movement speed between GPS logs.

The number of outings was calculated based on the number of times people went out per day (the number of times they stayed outside their homes), and the time spent out was calculated based on the sum of the time spent outside their homes plus the time spent traveling (end of day—start of day—time spent at home). To determine whether a person was “staying” or not, data on a person observed within a range of 100 m continuously for more than five minutes were judged to indicate “staying”, and the coordinates of the center of gravity within that time were considered to indicate “staying position”. As for the means of transportation, the number of outings according to means (walk: walking, cycle: bicycling, car: car and others) was calculated separately for weekdays and holidays in 2019 and 2020. Due to the difference in the total number of outings between 2019 and 2020, the change in the percentage of each mode of transportation was calculated.

#### 2.3.2. Percentage Increase/Decrease in the Number of Visits to Each Greenspace and Distance Traveled

The number of visits for each large greenspace was divided between 2019 and 2020. We also calculated the percentage increase or decrease based on the number of visits in 2019 and 2020 for each greenspace. We calculated the median access distance of visitors to each greenspace and examined the relationship between the rate of increase/decrease, the number of visits, and the distance traveled.

## 3. Results

### 3.1. General Change of Activity Pattern

To describe the overall pattern of movement change, as a pre-analysis, we visualized the movement of holiday-goers in Kanazawa in 2019 and 2020 from the collected GPS data (Figure 2a,b). Many people were seen visiting Kanazawa Station (the central transportation hub) and Kenrokuen Garden area (a major tourist destination with a representative feudal lord garden of the Edo period) in 2019 (Figure 2a), but few people were seen in 2020 (Figure 2b).

During the 2020 emergency declaration period, the paid section of Kenrokuen Garden was closed, and the parking lots of other large prefectural facilities were closed from the middle of the study period. However, entry to greenspaces other than Kenrokuen Garden was not prohibited (only social distancing measures were requested). As for public transportation, although service on the Shinkansen bullet train (direct access to Kanazawa Station) was reduced, buses were running more or less normally.

### 3.2. Average Number of Outings and Total Time Spent out of the House Per Day

A nested ANOVA result is shown in Table 1. “Type of day” (weekdays or holidays) is nested within “year.” Both factors are treated as fixed effects. The average number of times in which people went out on weekdays and on holidays decreased to 62% (weekdays) and 52% (holidays) from 2019 to 2020 (*p* < 0.001) (Figure 3). Furthermore, the average number of outings within the year is not equal between weekdays and holidays (*p* < 0.001). Similarly, comparing 2020 to 2019, the average total time spent out of the house decreased on both to 64% (weekdays) and 52% (holidays) (*p* < 0.001) (Figure 4), indicating that people avoided going out at all and for long periods in 2020. Within the year, the average total time spent out of the house per day is not equal between weekdays and holidays (*p* < 0.001) (Table 2, Figure 4).

### 3.3. Maximum Distance from Home to Place of Visit for a Day

Figure 5 shows a frequency distribution of the maximum distance travelled from home to the place of visit for a day. Distance from home to place of visit in both 2019 and 2020 commonly ranged between 0 and 1000 m. In 2020, a smaller percentage of people visited places located more than 1000 m from home than in 2019. The proportion of long-distance travel, particularly more than 4000 m, decreased in 2020 (Figure 5), indicating that people avoided long-distance travel in 2020.

### 3.4. Number of Outings by Means of Transportation

Outings by car and other means of transportation (other than walking and bicycling) accounted for the largest proportion in both 2019 and 2020. An increase in the percentage of outings by foot and a decrease in the percentage of outings by bicycle to 70% (weekdays) and 82% (holidays) of the 2019 level was recorded in 2020 (Figure 6). In 2020, a year-on-year increase of 110% was recorded in the percentage of outings by foot during both weekdays and holidays. The percentage of outings by car and other means decreased very slightly (99%) on holidays and increased slightly (104%) on weekdays.

### 3.5. Number of Visits by Destination

No significant change was recorded in the percentage of visits by destination for green areas including parks and commercial areas between 2019 and 2020. Here the greenspaces are large green areas (≥10,000 m^2^), and the commercial areas are shopping malls with several large stores. Year on year, the percentage of visits to commercial areas decreased to 77% and 91% on holidays and weekdays, respectively, in 2020 (Figure 7). However, for greenspaces, the percentage of visits increased very slightly on holidays and weekdays in 2020 (Figure 7). Thus, year on year, the percentage decrease for green areas was smaller (Figure 7), with even a slight increase of 100.06% during the consecutive holidays in May.

### 3.6. Percentage Increase/Decrease in the Number of Visits to Each Greenspace and Distance Traveled

While the number of visits to large-scale greenspaces in central Kanazawa (e.g., Kanazawa Castle Park and Ishikawa Shiko Memorial Park), which had many faraway visitors and were used more frequently in 2019, decreased significantly to half or less in 2020, the percentage of visitors who traveled no more than 1000 m increased in 2020 (Figure 8). In Figure 8, the horizontal axis shows the percentage change in the number of visits at a given greenspace in 2020 compared to 2019. The median of travel distances in 2019 and 2020 for a given greenspace is plotted vertically according to the corresponding percentage change in the number of visits. The vertical axis is the access distance from the user’s home, so a particular greenspace is plotted at the median distance in 2019 and 2020, respectively. Figure 8 shows that (1) the distance traveled in 2020 was generally shorter than in 2019 (2020 bubbles located below 2019 bubbles), and (2) greenspaces that experienced a large increase in the number of visitors in 2020 (greenspaces plotted on the right side of the graph) have a visiting distance of around 1000 m, which means that they are often visited by people from nearby areas.

## 4. Discussion

### 4.1. Number of Outings and Total Time Spent Outside the Home

The overall trend (Figure 2a,b), the analysis on the average number of outings per day (Table 1, Figure 3), and the average total time spent outside one’s home per day (Table 2, Figure 4) show that Kanazawa citizens went out less often both on weekdays and holidays in 2020 than in 2019. Furthermore, when they did go out, they spent less time outside the home both on weekdays and holidays. This was expected given that the data in 2020 coincided with an emergency declaration issued by both Ishikawa Prefecture and the national government. Even though the movement restrictions in Japan are a “request”, most people obeyed as the number of the newly infected rose rapidly [31,38].

Faced with movement restrictions, people in other countries also went out less often and reduced activities that could be considered non-essential or high-risk [9,39]. For example, Burnett et al. [40] showed that 63% of British adults surveyed reported spending less time visiting greenspaces following movement restrictions. Similarly, 67% of US college students limited their outdoor recreation activities to some extent, and 54% reduced their park use during the early stages of the COVID-19 pandemic in 2020 [13].

### 4.2. Maximum Distance from Home to Place of Visit and Number of Outings by Means of Transportation

We also investigated how patterns of outings changed. As for the maximum distance from home to place of visit, though between 0 and 1000 m was the most common visitation distance for both 2019 and 2020, 2020 saw a decrease in visitation distances of more than 1000 m, and rarely more than 4000 m even on holidays (Figure 5). These findings show that Kanazawa citizens visited places that were close (up to 1000 m) to home during the pandemic.

Given the decrease in the number of times people chose to go out and by car as the most popular mode of travel, the percentage of outings by foot increased, outings by bicycle decreased, and outings by car showed little change (Figure 6). The percentage of outings by foot increased year-on-year for holiday-goers in 2020. Therefore, we can infer that more people visited nearby places on foot in 2020, or it may be that more people chose to walk to the destination since they were close to their homes.

### 4.3. Number of Visits by Destination

The percentages of greenspace visits and commercial area visits did not change significantly between 2019 and 2020. Although the percentage of visits to shopping malls decreased in 2020 year on year, the percentage of visits to large greenspaces actually increased very slightly (Figure 7). Therefore, we can conclude that although people went out less often in 2020, they did not refrain from visiting green areas as much as commercial areas.

### 4.4. Percentage Increase/Decrease in the Number of Visits to Each Greenspace and Distance Traveled

Figure 8 revealed a detailed trend on visits to green spaces. First, as we have discussed, Kanazawa citizens generally visited greenspaces that were closer in 2020. Second, those greenspaces that received increased visits in 2020 tended to be visited by people living within 1000 m. The results indicate that while the number of visits to greenspaces and parks in the city center decreased, the number of visits to those parks and greenspaces that are close to residential areas increased. These results have implications for the importance of greenspaces in and near residential areas when lifestyle restrictions are in effect.

### 4.5. Overall Discussion

The results show that during the COVID-19 pandemic, people avoided going out, including visits to large parks in the city center, a popular destination in 2019, to presumably avoid crowding and unnecessary social contact. However, Figure 8 shows that greenspace visits within 1000 m of home increased during the pandemic, and visits to nearby greenspaces around residential areas increased in particular. We also found that long-distance (>4000 m) outings decreased (Figure 5) and that people tended to visit places within 1000 m of home on foot (Figure 6). Therefore, it can be inferred that nearby greenspaces within a 1000 m radius from home provided Kanazawa citizens with health maintenance and refreshment opportunities during the early stages of the COVID-19 pandemic.

These findings are corroborated by studies in other countries conducted over a similar time period. For example, Venter et al. [10] noted that access to greenspaces interwoven within the urban matrix in Oslo for outdoor recreational activity became as important within residential areas and nearby city parks as in the forested land-use zone during the partial lockdown. Ugolini et al. [9] found that people in Israel, Italy, and Spain tended to visit greenspaces at closer distances (<200 m) during the lockdown. In all countries (Croatia, Israel, Italy, Lithuania, Slovenia, and Spain) surveyed, the most common means of transportation used by urban greenspace visitors was their feet. Ugolini et al. [9] also pointed to the importance of diverse urban greenspace and related means of transportation: people walking to small urban gardens nearby (e.g., in Italy) or tree-lined streets (e.g., in Spain, Israel), and people traveling by car to green areas outside the city (e.g., in Lithuania). Similarly, Korpilo et al. [41] showed that Helsinki residents were more likely to visit greenspaces closer to their homes for recreation during the pandemic than before the pandemic. Residents were more likely to visit nearby residential areas with high tree cover density, underscoring the importance of urban forests and tree-rich parks in their neighborhoods.

Faced with movement restrictions, people were commonly observed to visit available greenspaces nearby. For example, in Italy, a reduction in visits to urban parks was combined with an increase in visits to gardens and other nearby greenspaces [39]. Similarly, in Iran, although visits to both public and private greenspaces decreased during the pandemic, the degree of decrease in visitation of private greenspaces such as gardens or courtyards by those with access was less pronounced [42]. Our study also found that although overall outings decreased, the degree of decrease in visitation of greenspaces was lower than for commercial areas (Figure 7). A study in Brisbane found that those with easily accessible greenspace nearby such as a backyard increased their greenspace use during the pandemic [43]. These studies suggest that people inherently felt a greater need to visit greenspaces for psychological respite perhaps due to movement restrictions [13,22,34,39,40].

Studies in Japan that used location data of smartphones to examine the effect of the COVID-19 pandemic on people’s greenspace use show similar trends as our study. For example, Takeyama et al. [31] indicated that various use restriction measures and social distancing effectively decreased the number of visitors during the pandemic for all large public parks being studied in Hyogo Prefecture. However, one park showed a particularly minor degree of reduction. They hypothesized that this was due to the fact that it was closer to urban areas, visited on foot by people who lived nearby. Amemiya et al. [32] indicated that the public parks in Tsukuba City in Ibaraki Prefecture that received more visitors during the pandemic in 2020 than in 2019 tend to be smaller in size (~4 ha) and that (1) the reduction in park visitors was much larger on holidays than on weekdays, and (2) for the same park on holidays, the number of users increased, especially in parks located near single-family homes. This supports our study’s finding that the number of visits to those parks and greenspaces that are close to residential areas increased in 2020.

Our study underscores the importance of greenspaces including parks that are close (≤1000 m) to people’s homes for mental and physical health maintenance in the context of COVID movement restrictions. While this finding is corroborated by some other studies, as for urban planning, it is not simply that more parks and other greenspaces close to people’s homes should be created to increase park use and health benefits. Our analysis also revealed that even if a greenspace is located within 1000 m from home, many people do not use it. Although the parks closest to home are those that are close to daily life and can be used easily, they are not always the most frequently used during normal times [44]. Diversity of greenspaces (area, ownership, tree coverage, etc.), distance from city center, and diverse facilities that meet the demands of park users are also important factors in people’s decision to visit a certain park and other greenspace [11,12,13,34,38,39,45,46]. Organizing activities and events, policies, and education to increase people’s emotional affinity with nature is also important for encouraging people’s use of and visits to greenspaces [47,48].

### 4.6. Study Limitations

The human flow data used in the analysis have some limitations. First, the results were affected by the spatial accuracy of the data, which have an error accuracy of about 50 to 100 m. To avoid this problem, we limited our analysis to large greenspaces and assumed that those who “stayed” near these greenspaces were users or beneficiaries of the greenspaces. However, we may have overestimated the extent to which people who live in houses bordering greenspaces are greenspace users. Second, the period covered by the data, from May 1 to 15, which includes consecutive holidays, may be neither a representative sample of a typical year nor a good sample for examining the behavioral changes of Kanazawa citizens. Although these criticisms are valid, precisely due to the fact that more people tend to travel long distances or go outside during the selected period, we may be able to examine stark changes in people’s outing behaviors before and during the first wave of the COVID-19 pandemic. Third, we were not allowed access to the demographic information on the smartphone holders. Since we were able to track down movements and places visited by a specific smartphone holder, the personal information privacy law of Japan prohibits linking this spatio-temporal data to any personally identifiable information. This limitation forbade associating outing behavioral changes to certain demographic information (e.g., age, gender, income, occupation) and conducting analysis as in other studies [13,40,49]. Had this kind of analysis been possible, we might have been able to provide more social justice-related suggestions concerning the use and location of urban greenspaces and certain socially vulnerable groups such as elderly or low-income populations.

## 5. Conclusions

The results of this study’s analysis indicated changes in the behavior of Kanazawa citizens around outings before and during the declaration of a state of emergency under the COVID-19 pandemic in Japan in April–May 2020. In sum, Kanazawa citizens strongly tended to avoid going out, with a decrease in the number, time, and distance of outings. However, while the citizens refrained from going out, the rate of decrease in the number of visits to greenspaces was smaller than for commercial areas, and the number of walks increased after 2019. The number of visits to nearby greenspaces increased slightly compared to 2019, while people tended to avoid using greenspaces in the city center. These results suggest that those who had greenspaces around their homes were able to use them to maintain their health, refreshment, and other purposes during the COVID-19 pandemic, according to the human flow big data. However, those who did not have greenspaces nearby went out less frequently but did not show an increased trend of staying at home.

Urban parks and greenspaces are examples of green infrastructure that provide a variety of benefits to urban residents. Through their ecological functions, they can also offer solutions to societal challenges in cities (i.e., nature-based solutions). Historically, urban parks were developed in the eighteenth and nineteenth centuries to ensure public health and reduce environmental pollution [50]. In recent years, it has been reported that greenspaces have various functions such as disaster prevention (e.g., shelter and refuge in case of disasters), microclimate mitigation, aesthetic pleasure, branding of the area, increase in land prices in the surrounding area, habitats for local organisms, and places for local community activities. However, cities differ greatly worldwide in the amount of greenspace around residential areas, and the more affluent districts tend to have more developed greenspace [51]. Under the Sustainable Development Goals, however, agreed upon by the United Nations to be achieved by 2030, Goal 11 states, “Make cities and human settlements inclusive, safe, resilient and sustainable” with target 11.7: “By 2030, provide universal access to safe, inclusive and accessible, green and public spaces, in particular for women and children, older persons, and persons with disabilities”. This study, conducted during the COVID-19 pandemic, reiterates the need for green infrastructure that is closely related to people’s lives, such as urban greenspaces, and shows the importance of creating a society in which all urban residents, regardless of wealth, can enjoy the benefits of greenspaces through strategic urban planning.

## Figures and Tables

**Figure 1 ijerph-19-08728-f001:**
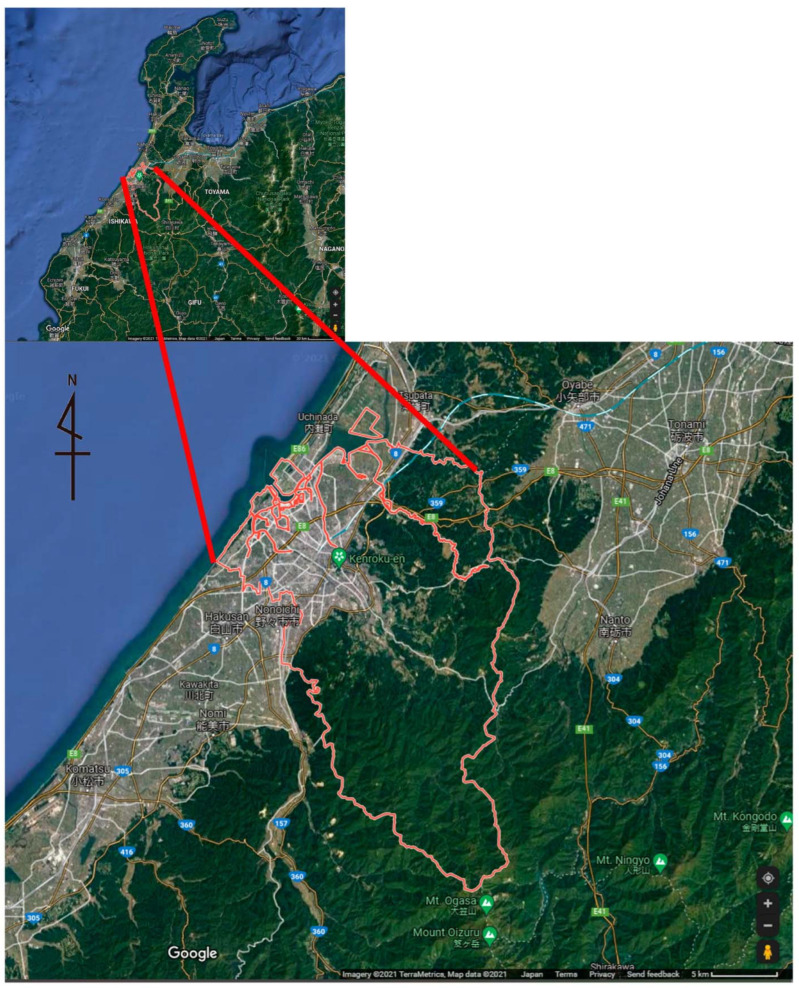
Location of Kanazawa City in Ishikawa Prefecture, Japan.

**Figure 2 ijerph-19-08728-f002:**
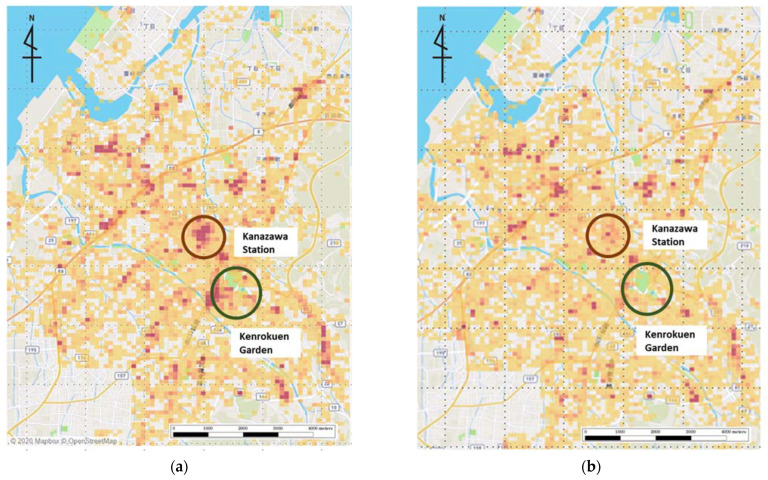
(**a**) (**left**) and (**b**) (**right**) (**a**) Movement of people on holidays in 2019. The darker the color, the more people were concentrated in the area. (**b**) Movement of people on holidays in 2020. The darker the color, the more people were concentrated in the area.

**Figure 3 ijerph-19-08728-f003:**
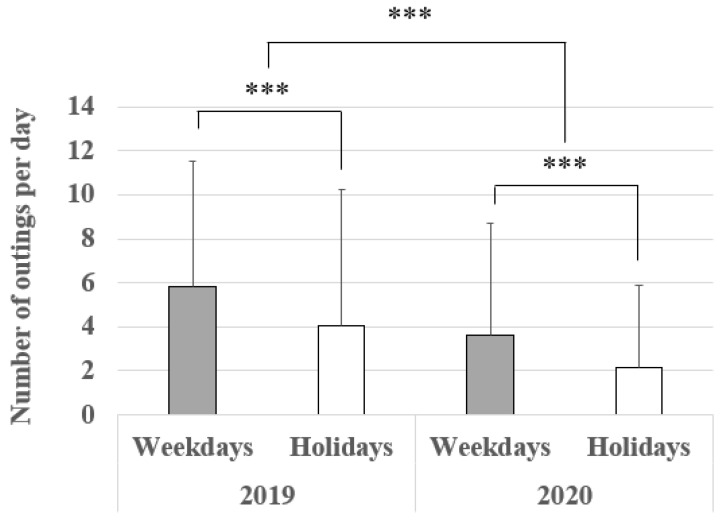
Number of outings per day (mean ± S.D). *** designates *p* < 0.001. *n* (2019) = 55,713 and *n* (2020) = 42,197.

**Figure 4 ijerph-19-08728-f004:**
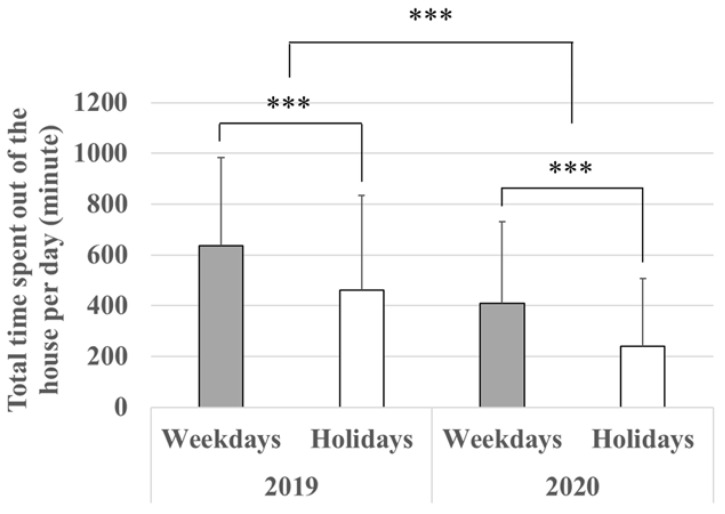
Total time spent out of the house per day (mean ± S.D). *** designates *p* < 0.001. *n* (2019) = 38,011 and *n* (2020) = 42,197.

**Figure 5 ijerph-19-08728-f005:**
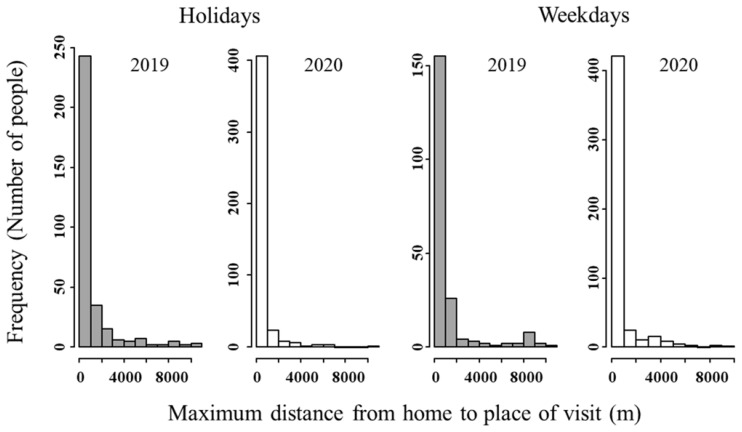
Maximum distance from home to place of visit for a day. Note that the vertical scale differs between 2019 and 2020.

**Figure 6 ijerph-19-08728-f006:**
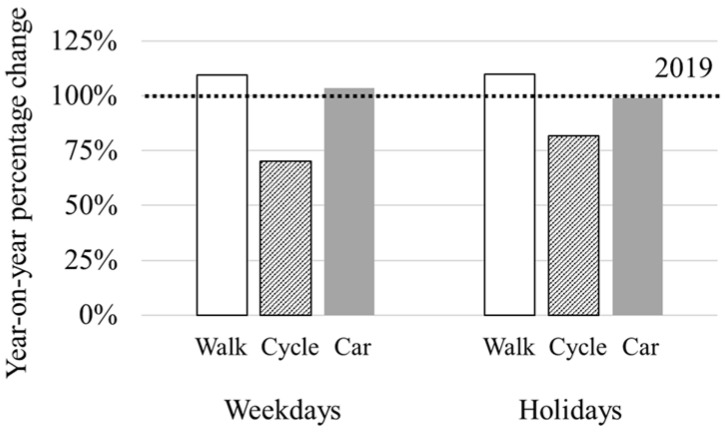
Percentage change by means of travel in 2020 compared to 2019.

**Figure 7 ijerph-19-08728-f007:**
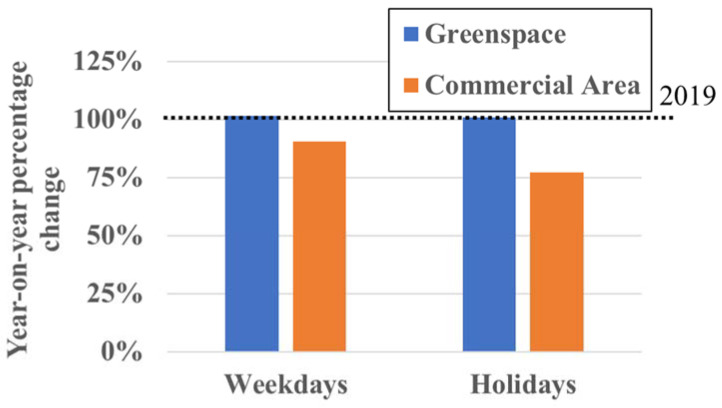
Percentage change by destination in 2020 compared to 2019.

**Figure 8 ijerph-19-08728-f008:**
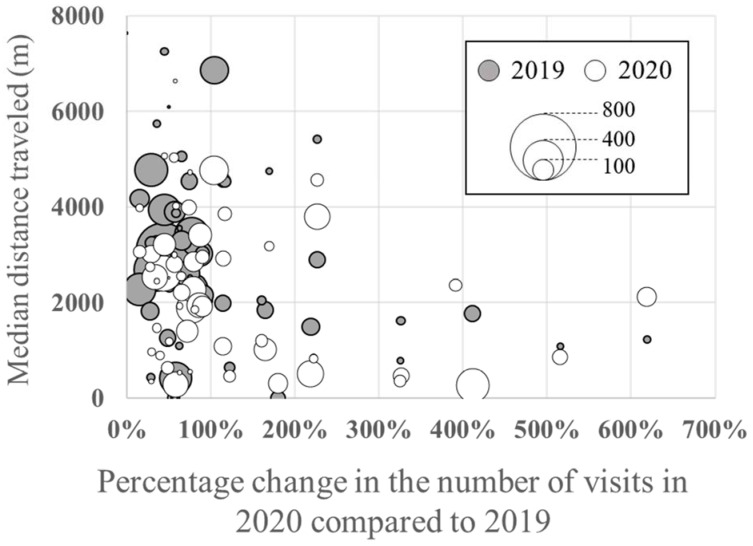
Relationship between the rate of increase/decrease in the number of visits to each greenspace in Kanazawa City in 2020 compared to 2019 and the average distance traveled from home. Note that a bubble size indicates the number of visits.

**Table 1 ijerph-19-08728-t001:** Nested ANOVA of number of outings per day.

Source	df	Sum Sq	Mean Sq	F Value	*p* Value
Year	1	69,679	69,679	2648	<2 × 10^−16^ ***
Type of Day (Year)	1	50,299	50,299	1911	<2 × 10^−16^ ***
Residuals	80,205	2,110,650	26		

*** Designates *p* < 0.001.

**Table 2 ijerph-19-08728-t002:** Nested ANOVA of total time spent out of the house per day.

Source	df	Sum Sq	Mean Sq	F Value	*p* Value
Year	1	803,900,000	803,873,362	7335	<2 × 10^−16^ ***
Type of Day (Year)	1	587,100,000	587,103,496	5357	<2 × 10^−16^ ***
Residuals	80,205	8,790,000,000	109,591		

*** Designates *p* < 0.001.

## Data Availability

Data was obtained from Blogwatcher Inc. and are available from Y.U. with the permission of Blogwatcher Inc.

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
