# Peer review of "Human Flow Dataset Reveals Changes in Citizens’ Outing Behaviors including Greenspace Visits before and during the First Wave of the COVID-19 Pandemic in Kanazawa, Japan"

_ijerph, 2022, doi:10.3390/ijerph19148728_

Round 1
Reviewer 1 Report
- In the introduction, the authors should avoid just listing references. The findings of the different references should be at least introduced.
- For example: The COVID-19 pan- 36 demic has also highlighted the important health benefits of urban greenspaces (Rojas- 37 Rueda et al 2019; Grima et al 2020; Lesser and Nienhuis, 2020; Ugolini et al 2020; Venter et 38 al 2020, 2021; Lu et al 2021; Larson et al 2022). -> what did they all say are the benefits?
- Another example is: Few studies have used human flow data (mobile phone location data) to examine 71 people’s greenspace use (Amemiya et al 2020; Hu et al 2020; Takeyama et al 2021). -> If this is your contribution you need to be clear about what they did and how you’re doing something different and filling a gap.
- The introduction / literature review should also expand on research about the use of parks and green spaces, not just on the benefits and then on COVID measures. Also, other studies on the use of public spaces during COVID need to be discussed.
- The aim of the paper is unclear. What is the research questions or hypothesis the authors are addressing?
- It seemed it was about access to parks and greenspace, and then in the analysis (which is more descriptive than analytical) several other elements are included and not much discussion on access to parks and greenspace is found.
- Is the methodology the only innovation and significance of this article?
- Why was this particular case study selected? It is not clear. Also, if May has so many public holidays, why was it selected?
- Figure 5 is not clear - the analysis is also insufficient.
- The limitations of the study are clear and well-developed.
- Overall, the paper is ok, but the relevance needs to be strengthened. Also, to make it really valuable, data from 2021 should be included to see if there were changes post-restrictions. Finally, cultural nuances should be included in the discussion and conclusions.
Author Response
>In the introduction, the authors should avoid just listing references. The findings of the different references should be at least introduced.
Response: We added a couple of examples. The content of the first paragraph in the introduction is well established in past research, and relevant literature was cited in appropriate places. The literature that studied the effects of greenspaces on health is referred to again in the discussion (section 4).
>The introduction / literature review should also expand on research about the use of parks and green spaces, not just on the benefits and then on COVID measures. Also, other studies on the use of public spaces during COVID need to be discussed.
Response: The previous paper title and the study aims were misleading. Therefore, we changed them to more accurately describe the study.
The study was intended to explore the changes in outing behaviors, NOT actual use of parks and other greenspaces. We changed the way the study aims are written. The introduction/literature review section should now be consistent with the aims. Examples were added that showed the health benefits of greenspaces, which is the functional focus of greenspaces of this paper.
>The aim of the paper is unclear. What is the research questions or hypothesis the authors are addressing?
It seemed it was about access to parks and greenspace, and then in the analysis (which is more descriptive than analytical) several other elements are included and not much discussion on access to parks and greenspace is found.
Response: We were interested in investigating how citizens' outing behaviors may have changed before and during the first wave of the COVID-19 pandemic in Kanazawa, Japan.
The aims of the study were vague. We revised the study aims so that they were written more clearly and in detail. Now the aims should be consistent with the results and discussion. The title of the paper was also changed to more clearly describe the study aims.
>Is the methodology the only innovation and significance of this article?
Response: The use of mobile phone location big data to examine the changes in Kanazawa citizens' outing behaviors and visitation patterns, and visits related to large greenspaces, is certainly a unique characteristic of the study. We are not stating that this is the only significance of the study. We believe that the contribution of the study is well shown in the results and discussion.
>Why was this particular case study selected? It is not clear. Also, if May has so many public holidays, why was it selected?
Response: Since the aim of the study is to compare changes in people's outing behaviors before and during the pandemic, and many people tend to travel (go outside) during the long holiday week in May, we hypothesized that large differences in outing behaviors could be observed in the selected study period (May 1-15). This reasoning was added to section 2.2.
>Figure 5 is not clear - the analysis is also insufficient.
Response: Further explanation about Figure 5 was added in the results.
>Overall, the paper is ok, but the relevance needs to be strengthened. Also, to make it really valuable, data from 2021 should be included to see if there were changes post-restrictions.
Response: Thank you for your suggestions.
We revised the paper with more examples and discussion, so the relevance could be strengthened.
We compared 2020 data, in the middle of the first wave of the global COVID-19 pandemic, with 2019 data. This comparison highlights the stark difference between before and during the pandemic.
>Finally, cultural nuances should be included in the discussion and conclusions.
Response: A complete assessment of the effects of culture (including customs and behavioral norms and religion) is beyond the scope of this paper. Nonetheless, Japan's request-based style of societal measures taken to prevent the spread of COVID-19 and how people and businesses complied were already described in the introduction and were included in the discussion (originally in lines 240-242).
Reviewer 2 Report
This article tries to reveal changes in the use of urban parks and other greenspaces in May 2019 and May 2020 in Kanazawa, Japan with human flow data. It is topical and would be inspiring for urban greenspace development in future. However, the authors need to make the contents more focused and deepen the discussion. Please see my comments and suggestions as below:
1. Data presented in sections 3.2 to 3.4 are off track from the topic chosen (seemingly target of this article as stated in lines 74-76). They do not directly reveal visit patterns related to urban parks and other greenspaces. Meanwhile, actual patterns of urban parks and greenspaces visits are not studied specifically and thoroughly with source data.
2. The authors should clarify any impact on operations and availability of greenspaces and public transportations by the pandemic, i.e., any greenspace closed down or any public transportation stopped or reduced service during the studied periods? These may also affect people's behavior.
3. Some parts of contents are repeating, and the authors are suggested to merge them into proper sections. For instance, contents in lines 49-57 are more suitable to merge with lines 94-102; contents in lines 68-70 should be part of Methods; sections 4.1-4.4 can be merged into relevant parts in the results section or be discussed in-depth.
4. The authors should disclose sample size for 2019 and 2020 separately. If the sample sizes of the two years are different, percentage of behavioral changes between 2019 and 2020 would be more reasonable instead of definite number of people presented currently.
Author Response
This article tries to reveal changes in the use of urban parks and other greenspaces in May 2019 and May 2020 in Kanazawa, Japan with human flow data. It is topical and would be inspiring for urban greenspace development in future. However, the authors need to make the contents more focused and deepen the discussion.
Response: Thank you for your comments and suggestions. With the changes and additional information (please see below), we believe that the revised manuscript shows changes in the overall visiting behaviors of Kanazawa citizens and those specific to greenspaces more clearly through a focused discussion of the results.
>1. Data presented in sections 3.2 to 3.4 are off track from the topic chosen (seemingly target of this article as stated in lines 74-76). They do not directly reveal visit patterns related to urban parks and other greenspaces. Meanwhile, actual patterns of urban parks and greenspaces visits are not studied specifically and thoroughly with source data.
Response:
We revised the aims of the study so that they match the presented results and following analysis.
Using human flow big data, we explored how citizens' outing behaviors may have changed before and during the first wave of the COVID-19 pandemic in Kanazawa, Japan, in April–May 2020. We first examined overall outing and visitation patterns related to urban parks and other greenspaces and then examined citizens' greenspace visits in relation to their travel distance from home.
>2. The authors should clarify any impact on operations and availability of greenspaces and public transportations by the pandemic, i.e., any greenspace closed down or any public transportation stopped or reduced service during the studied periods? These may also affect people's behavior.
Response: Thank you for the suggestion. We added discussion of minor effects of the operations and public transportation on people's outing behaviors and access to parks and other greenspaces in section 3.1. Only the paid section of Kenrokuen Garden was closed down during the pandemic.
>3. Some parts of contents are repeating, and the authors are suggested to merge them into proper sections. For instance, contents in lines 49-57 are more suitable to merge with lines 94-102; contents in lines 68-70 should be part of Methods; sections 4.1-4.4 can be merged into relevant parts in the results section or be discussed in-depth.
Response:
The content in lines 49-57 is needed to state the difference between countries in the degree of movement restrictions and how Japan as a whole dealt with the pandemic. Meanwhile, the content in lines 94-102 needs to stay in section 2.1, which describes the study area (Kanazawa City in Ishikawa Prefecture). Lines 94-102 described the COVID-19 circumstances in Ishikawa Prefecture where the study area is situated.
The content in lines 68-70 was moved to a better location to make the argument flow better.
More discussion was added to sections 4.1-4.4. We believe that what we have (sections 4.1-4.4) is a reasonable interpretation and discussion of the corresponding results. These sections are also necessary as background knowledge for section 4.5 and should be left here instead of being merged with the results.
>4. The authors should disclose sample size for 2019 and 2020 separately. If the sample sizes of the two years are different, percentage of behavioral changes between 2019 and 2020 would be more reasonable instead of definite number of people presented currently.
Response: The sample size for 2019 and 2020 was added in the figure captions (Fig. 3 and Fig. 4). Figures 3 and 4 show the average number of the very large sample, so that the effect of sample size difference should be minimal. Figures 6 and 7 already show the percentage change of each travel mode and large greenspace/shopping mall visit, and this is stated in the methods, section 2.3.
Reviewer 3 Report
I really enjoyed reading your paper, thank-you. It provides a clear contribution to emerging knowledge around societal, behavioural changes during the global pandemic, particularly around the use of green-spaces. As you say possibly the key contribution is methodological in the use of GPS based big-flow data which I found to be exemplary with the acknowledgement of one or two limitations to the mapping possible. I also thought that the analysis of data was extremely rigorous and appropriate. Personally I would have liked to have seen the data backed up by some additional qualitative data provided by an anonymous questionnaire to your participants but maybe that is for a future study. I will be recommending publication following a final spell-check as I believe there are one or two small grammatical errors which should be corrected.
Author Response
Thank you for your time in reviewing our paper. We appreciate your favorable comments and recommendation for publication. We edited the paper based on other reviewers' comments.
Round 2
Reviewer 2 Report
The article has been much improved and can be accepted for publication.